# Light Scattering from Rough Silver Surfaces: Modeling of Absorption Loss Measurements

**DOI:** 10.3390/nano11010113

**Published:** 2021-01-06

**Authors:** Matin Dehghani, Christin David

**Affiliations:** 1Department of Physics, University of Kashan, 8731753153 Kashan, Iran; matin99dehghani@gmail.com; 2Institute of Condensed Matter Theory and Optics, Abbe Center of Photonics, Friedrich-Schiller-Universität Jena, 07743 Jena, Germany

**Keywords:** nanotextured surfaces, diffuse scattering, theory and simulation, rough surface morphologies

## Abstract

We consider two series of experimental setups of multilayered Ag/ZnO thin films with varying surface morphologies given by atomic force microscopy images. The absorption loss under diffuse scattering is studied theoretically by applying a combination of the scattering matrix approach with diffraction theory for randomly nanotextured interfaces. Our modeling is in excellent agreement with the respective measurements. The theoretical approach is applicable to a wide range of wavelengths, surface morphologies, and materials for both measured and computed rough surface morphologies.

## 1. Introduction

Regular nanostructures, such as diffraction gratings [1], photonic crystals [2,3], and metasurfaces [4,5], allow tuning of their optical response to the desired spectral properties, e.g., in nanoparticle and hole arrays [6,7,8,9,10], as well as their near-field patterns under varying illumination conditions, e.g., for optical data storage [11,12]. Fabrication techniques range from chemical [13] to physical methods, such as self-assembly [14], laser-induced [15], and annealing [16,17] procedures. This allows a high geometrical control over the produced structures.

Photocatalytic [18,19,20], photovoltaic [21,22], and some biochemical sensing applications, such as Raman spectroscopy [3,23,24], on the other hand, rely on large surface or contact areas. In such systems, strong local field enhancement or enhanced forward scattering across the entire surface is desired. Likewise, a broad spectral response, rather than specific spectral lines created at distinct hot-spot positions, is of importance. Rough surface morphologies address this need and are, thus, of high interest in a number of such applications. Typically, they are easier and cheaper to produce, which makes them important in large-scale industrial applications, such as solar [25,26] and fuel cells compatible with mass production lines [27]. Moreover, in realistic setups, surface roughness can often not be neglected, and diffuse scattering with possibly inherent additional losses should be taken into account when considering the optical response of any nanostructure.

In this article, we consider the two Ag/ZnO samples depicted in Figure 1, which were investigated experimentally with photothermal deflection spectroscopy by Springer et al., Ref. [28], as potential back reflectors for solar cells. The experimental setups consisted of a 500 nm ZnO film on a glass substrate supporting a silver layer with varying surface roughness in a CCl${}_{4}$ environment. These comprise the first sample type in Figure 1. In a second step, an additional ZnO film of 30 nm was coated onto the original samples to comprise a second batch, which we denote as sample type 2 in accordance with the original experimental study [28]. Experimentally, an increase in absorption was observed, which was related to the surface plasmon absorption.

To simulate the absorption loss in these systems, we employ a theoretical model for light scattering from rough and nanotextured surfaces [29,30,31,32], which combines diffraction theory for large-scale irregular surfaces with the scattering matrix approach for planar multilayered systems [33,34]. The next section explains this approach in detail and applies it to the original atomic force microscopy (AFM) images of six different surface morphologies provided by Springer et al. [28]. Table 1 lists the root mean square (rms) roughness values for the samples used. Our simulations show excellent agreement with the measured absorption loss. This approach enables us to consider further material combinations for the fabricated samples, as well as arbitrary surface morphologies beyond the ones provided experimentally.

## 2. Methodology

The frequency-dependent permittivity function
(1)$$\begin{array}{c} \epsilon \left(\omega \right)=\epsilon_{b}-\frac{\gamma \sigma }{\omega (\omega +i\gamma )}+\sum_{j=1}^{L}\left(\frac{i\sigma_{j}}{\omega -\Sigma_{j}}+\frac{i\sigma_{j}^{*}}{\omega +\Sigma_{j}^{*}}\right) \end{array}$$
is modeled based on the Drude–Lorentz theory with the large frequency (background) permittivity $\epsilon_{b}$, plasmonic damping rate γ, the electron conductivity σ, generalized conductivities $\sigma_{j}$ of bound electrons, and their respective Lorentz pole resonances $\Omega_{j}$. These parameters are taken from numerical fits to experimental data [35] for silver and other metals.

The basis for the description of the rough surfaces comprises AFM data spanning 10μm×10μm in physical size. Figure 2a shows three of the six available data sets with their respective height ranges. The elements of the height function z(x,y) for each surface are related to the pixel values in the grayscale image and are related to the height range in nanometers using the AFM data. For instance, the first surface in Figure 2a has a maximum height value of 80 nm (255 in grayscale), and the minimum height value is −52 nm (0 in grayscale). The AFM images thus define the different height functions z(x,y) for the investigated setups.

Following Refs. [31,32], we calculate the reflected (ν=R) and transmitted (ν=T) scalar electric fields $U_{\nu }$, which allow analysis of the total reflection and transmission in terms of specular and diffuse scattering. Hereby, the diffuse scattering contribution is given by the haze function $H_{\nu }$ and can be considered separately. It is defined as the (normalized) difference between the total reflection or transmission and the specular contribution (here at normal incidence):(2)$$\begin{array}{cc} H_{\nu }\left(\lambda \right) & =\frac{\nu_{\mathrm{tot}}-\nu_{\mathrm{spec}}}{\nu_{\mathrm{tot}}}. \end{array}$$

The scalar fields are
(3)$$\begin{array}{cc} U_{\nu }\left(k_{x},k_{y}\right) & =\frac{1}{2\pi }\int \int_{R^{2}}G_{\nu }\left(x,y\right)e^{-i(k_{x}x+k_{y}y)}dxdy, \end{array}$$
using the pupil functions
(4)$$\begin{array}{cc} G_{T}\left(x,y\right) & =\sqrt{\frac{T_{\mathrm{flat}}}{N}}e^{ik_{0}z\left(x,y\right)\left(n_{R}-n_{T}\right)},\\ G_{R}\left(x,y\right) & =\sqrt{\frac{R_{\mathrm{flat}}}{N}}e^{ik_{0}z\left(x,y\right)2n_{R}}, \end{array}$$
in two-dimensional *k*-space depending on the total transmission $T_{\mathrm{flat}}$ and reflection $R_{\mathrm{flat}}$ of the flat geometry. Hereby, $n_{R}$ is the refractive index of either the CCl${}_{4}$ environment ($n_{R}=1.4$ for type 1 samples) or the additional ZnO layer ($n_{R}=2.1$ for sample type 2), and $n_{T}$ is the refractive index of the silver layer. Hence, the rough interface is always the top of the silver layer. Using fast Fourier transform (FFT) analysis, the results of these integrals are depicted in Figure 2b for $U_{R}$ for three data sets, indicating the plethora of diffraction orders and scattering angles relevant to describing the optical response from such surfaces as compared to regular structures. Rather than as a continuous integral, the height function is defined via *N* pixels in the image, which is used here as a normalization. Hence, $k_{x}$ and $k_{y}$ vector components for each diffracted beam can be written in terms of the pixel position as $\frac{2\pi }{m}P$, in which *P* is a number from 0 to m−1 and m×m=N is the dimension of the image (256×256 pixels). The free-space wavenumber is $k_{0}=MM2\pi \;/\;\lambda$, so that $k_{0}^{2}n_{\nu }^{2}=k_{x}^{2}+k_{y}^{2}+k_{z}^{2}$ always. The red circle indicates where $k_{x}^{2}+k_{y}^{2}=k_{0}^{2}n_{\nu }^{2}$, and thus marks the border between regimes where $k_{z}$ becomes real (propagating modes for $k_{0}^{2}n_{\nu }^{2}-k_{x}^{2}-k_{y}^{2}>0$) or imaginary (evanescent modes for $k_{0}^{2}n_{\nu }^{2}-k_{x}^{2}-k_{y}^{2}<0$). Evanescent contributions lie, thus, outside of this circle, and are excluded in the following considerations, which is indicated by the truncated sum.

As seen in Equation (Equation 4), the total reflection $R_{\mathrm{flat}}$ and transmission $T_{\mathrm{flat}}$ of the flat interface are needed as inputs. $T_{\mathrm{flat}}$ and $R_{\mathrm{flat}}$ are calculated in a first step for the planar multilayer setup using the scattering matrix method [33,34]; i.e., the ideal planar structures from Figure 1a are considered, yielding the specular reflection and zeroth-order transmission values. Afterwards, the randomly nanotextured interfaces are accounted for. The deviation from the ideal result is then calculated by effective redistribution for diffuse scattering angles.

First, our simulation model processes the AFM images of the rough surface topography and extracts their two-dimensional height functions z(x,y) and rms roughness. We evaluated the rms roughness with minor deviations from the original work (see Table 2). In the numerical analysis of the AFM samples, some defects were cut from the input images of the height function to obtain surface roughness parameters close to the original evaluation of Ref. [28]. For this purpose, we cropped the affected AFM images and then resized them to the original numbers of rows and columns (256×256) using a bicubic interpolation algorithm. Furthermore, the obtained rms roughness was compared using Gwyddion [36], which specializes in data visualization and analysis for scanning probe microscopy images. Second, we then calculated the pupil functions $G_{\nu }\left(x,y\right)$ and two-dimensional Fourier transforms of each image, which led to the electric scalar fields $U_{\nu }\left(k_{x},k_{y}\right)$ for reflected and transmitted light.

It should be noted that the diffraction theory incorporated here is based on the paraxial approximation, and the diffraction at the rough interfaces results in a phase shift only. Both these restrictions apply when nanosized roughness is considered, but may break down for larger rms roughness. Moreover, we considered a single rough interface contributing to diffuse scattering, and no pathway of actual scattered beams is tracked, i.e., there is no interference between diffusely scattered beams, since the multilayer result of the flat reference enters the calculation. A more advanced approach has to improve on these aspects.

The total optical response, i.e., including diffuse and specular contributions to the reflection (ν=R) and transmission (ν=T), is determined from summing up all contributions within the unit circle (depicted in red in Figure 2b) where propagating modes are found:(5)$$\begin{array}{c} \nu =\sum_{k_{x}^{2}+k_{y}^{2}\leq k_{0}^{2}n_{\nu }^{2}}{\left|U_{\nu }\left(k_{x},k_{y}\right)\right|}^{2}, \end{array}$$
from which the absorptance is calculated in the usual way:(6)$$\begin{array}{c} A=1-T-R. \end{array}$$

The absorption loss as a function of the rms roughness is the main quantity considered in this work.

With these considerations, the diffuse scattering contribution, i.e., the haze function, results in
(7)$$\begin{array}{cc} H_{\nu }\left(\lambda \right) & =\frac{\sum_{k_{x}^{2}+k_{y}^{2}\leq k_{0}^{2}n_{\nu }^{2}}|U_{\nu }\left(k_{x},k_{y}\right){|}^{2}-{\left|U_{\nu }\left(0,0\right)\right|}^{2}}{\sum_{k_{x}^{2}+k_{y}^{2}\leq k_{0}^{2}n_{\nu }^{2}}{\left|U_{\nu }\left(k_{x},k_{y}\right)\right|}^{2}}. \end{array}$$

## 3. Results and Discussion

The haze in transmission for different surface morphologies is depicted in Figure 3 for the sample type 2 results. As can be expected, the largest haze is found for the highest rms roughness. It dominates the optical response over the specular contribution. The haze follows the overall form discussed by Jäger et al. [32]; however, the haze in transmission is strongly influenced by the additional metal film in the layered system and its surface plasmon polariton. A noticeable reduction in the transmission is seen around this resonance at about 315 nm, where the light is rather absorbed than transmitted. Furthermore, note the clear sequence of the samples: With increasing rms roughness, the haze strongly increases, almost reaching its maximum for the three largest rms roughnesses over the entire spectrum.

The overall absorption is shown in Figure 4 for the different rms values. Another important input value to consider is the layer thickness. Due to the rough morphology, the layer thickness of the central Ag layer is not a well-defined quantity. In the discussed experimental setup [28], the target thickness for the silver layer is 500 nm before increasing the surface roughness. However, this leaves an uncertainty about the final average layer thickness.

In the presented results, we have taken some liberty regarding the Ag layer thickness to find the best fit of the absorption loss data at different roughnesses (see Table 2). However, keeping the Ag layer film thickness constant for all sample types also well reproduces the measurements (see Figure 4b in comparison to Figure 4a). The final results are not altered by more than a percentage point. Overall, the curves are smoother, as the layers are too thick to show Fabry–Pérot-type oscillations, and absorption is larger as compared to the modified calculations, where we have substantially reduced the target thickness of the silver layer.

With the steps described above, we arrive at the absorption loss (shown in Figure 5 as a function of wavelength), which was calculated from Equations (Equation 5) and (Equation 6). The plasmon peak for silver is observed at a wavelength of about 315 nm (see the inset). Oscillations in the absorptance *A* with the wavelength, particularly for lower rms roughness, are due to light interferences in the thin film’s layered structure that has not yet been obscured by diffuse scattering from the rough surface. This is a direct result of the input of optical coefficients from the ideal flat structure, which accounts for multiple scattering effects.

In Figure 5a,b, we evaluate the absorption loss for the two sample types as a function of the rms roughness of the six samples at two distinct wavelengths. This is directly compared with the experimental results by Springer et al. [28], which are included as blue symbols. Excellent agreement with the measurements is found due to the inclusion of diffuse scattering.

In Figure 5c,d, a third-degree polynomial curve was fitted to the simulated absorptance for different rms to obtain a continuous function of the surface roughness for every wavelength between 200 to 1000 nm for the two sample types. The resulting contour plot allows the assessment of the absorption loss in the different layered structures, but also indicates a maximum of absorption for a specific rms roughness, as well as a best choice of rms roughness for broadband absorption. Hence, this simulation scheme allows specification of some characteristics of the samples for specific properties regarding the absorptance or transmission and reflection, respectively. Figure 5e zooms in on the plasmon peak, which remains stable for different values of rms roughness, at least within classical diffuse scattering theory. It could be expected that nano-scale features at the surface do not only contribute to diffuse scattering, but also to quantum phenomena, such as nonlocal optical response, particularly when considering metal surfaces [37,38,39]. This is already well understood with nanoparticles [40] and their impact on solar cells employing nanostructures [41,42], but not yet for rough surface morphologies, i.e., randomly sized and distributed features. Future work should consider such effects beyond classical diffraction optics.

The applicability of our modeling to further material combinations, different multilayer structures, and both experimental and numerical surface morphology samples is straightforward. Figure 6 shows, in analogy to Figure 5 for silver, the corresponding results if the silver film was replaced by gold or aluminum. First of all, it can be seen that the absorption loss with rough gold surfaces is similar to that of silver, with the difference that the plasmon peak is around 520 nm (see Figure 6c). Hence, the absorption is much larger over a broader spectral range, which can be more easily compared using spectrally integrated quantities, such as the overall photocurrent. In contrast, aluminum shows much larger absorption loss for the rms roughness given by AFM data, although its plasmon resonance lies in the ultraviolet range.

## 4. Conclusions

We investigated the optical properties of a multilayer containing a single rough Ag interface using scalar diffraction theory for randomly nanotextured thin films combined with the scattering matrix theory. First, the optical coefficients of the flat multilayer system are computed. Second, the surface of the top Ag layer is replaced with rough morphology data from AFM images using diffraction theory to calculate the diffuse scattering from these interfaces. With this, the absorption loss is calculated and compared to the measurements performed on the original samples.

The calculated absorption loss in Figure 5a,b agrees very well with the experimental results of Ref. [28].

This procedure can be applied to investigate any types of rough surface morphologies provided from fabricated samples via AFM or scanning electron microscopy (SEM) images or from numerically produced samples. It also enables us to investigate the effects of various shapes of nanoparticles and nanostructured surfaces using grayscale images for the height function within the approximations discussed earlier. This could be an interesting path towards the quick assessment of amorphous structures or smooth boundaries that are not accessible with standard tools.

## Figures and Tables

**Figure 1 nanomaterials-11-00113-f001:**
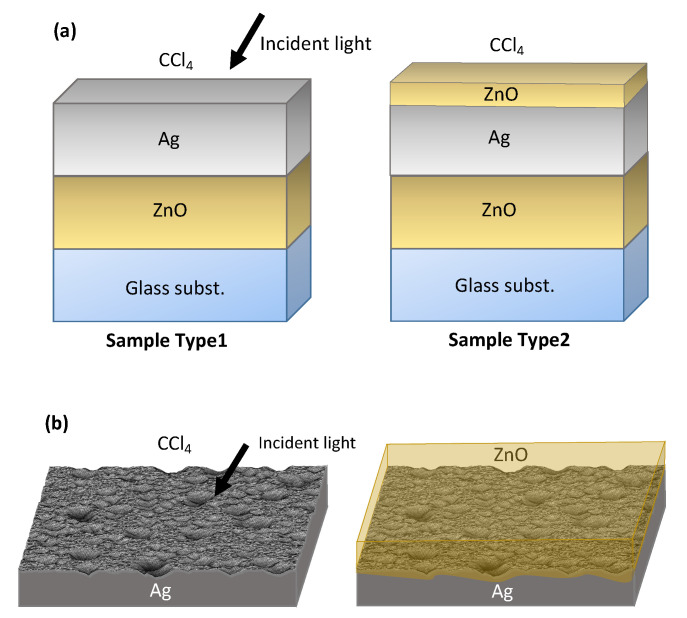
Illustration of the two series of Ag/ZnO setups considered. (**a**) Multilayered structures without rough surfaces for planar reference calculations and (**b**) their corresponding rough top interfaces, where diffraction theory is used in combination with scattering matrix calculations.

**Figure 2 nanomaterials-11-00113-f002:**
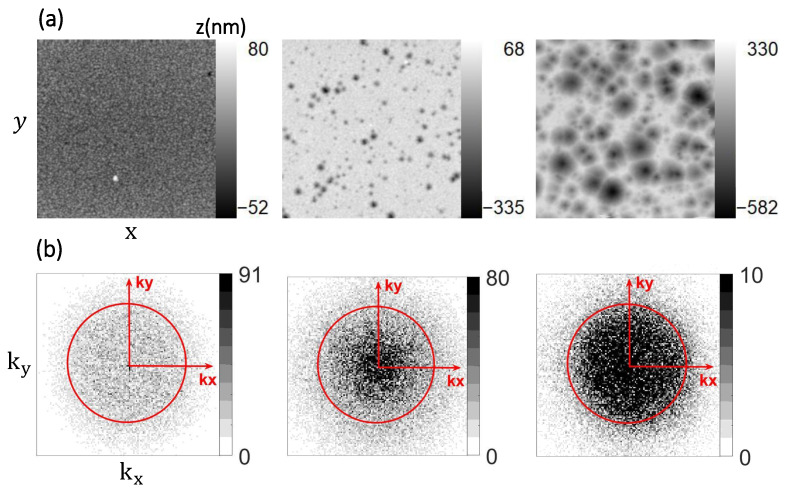
(**a**) Height function z(x,y) extracted from 10μm×10μm atomic force microscopy (AFM) images of rough Ag surfaces (Ref. [28]) with varying root mean square (rms) roughnesses of 8.86, 30.48, and 134.17 nm from left to right, respectively, and (**b**) the corresponding absolute values of their 2D Fourier components for the reflected scalar field (Equation (Equation 3) at 650 nm of wavelength). The red circle indicates the condition $k_{x}^{2}+k_{y}^{2}=k_{0}^{2}n_{1}^{2}$. Absolute values were scaled by a factor of 5 for better contrast, and images were resized to 128×128 pixels for clarity.

**Figure 3 nanomaterials-11-00113-f003:**
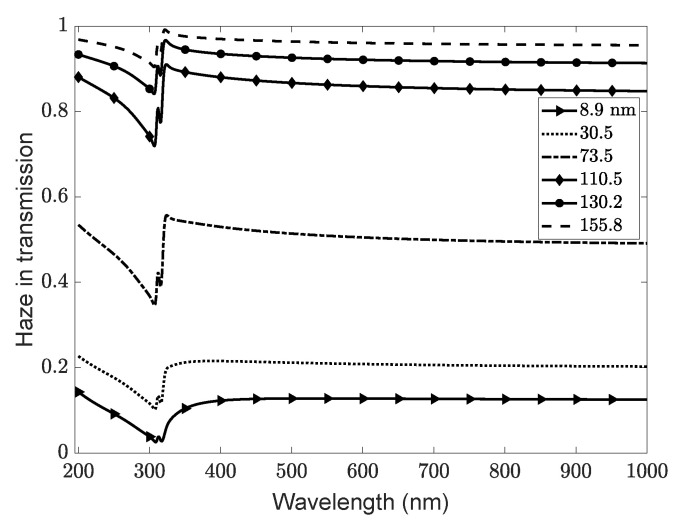
The (normalized) contribution of diffuse scattering is given by the haze, here in transmission $H_{T}=MM\left(T_{\mathrm{tot}}-T_{\mathrm{spec}}\right)\;/\;T_{\mathrm{tot}}$, for different morphologies and rms roughnesses of the ZnO/Ag interface of sample type 2. The silver plasmon peak is observed around 315 nm.

**Figure 4 nanomaterials-11-00113-f004:**
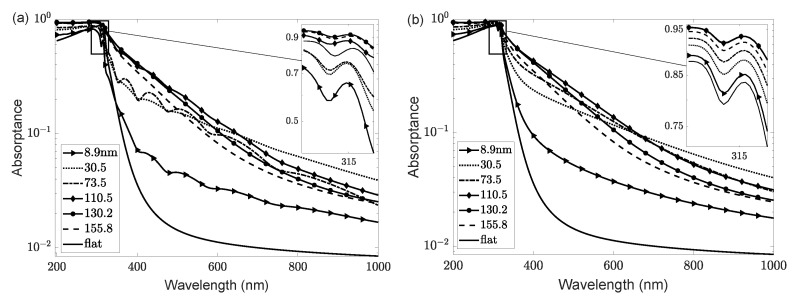
(**a**) Absorptance *A* as a function of wavelength for sample type 2 with various rms roughnesses of the ZnO/Ag interface on a logarithmic scale, including the ideal flat surface. The inset shows the silver plasmon resonance around 315 nm. (**b**) Same as (**a**) for constant Ag layer thickness of 500 nm.

**Figure 5 nanomaterials-11-00113-f005:**
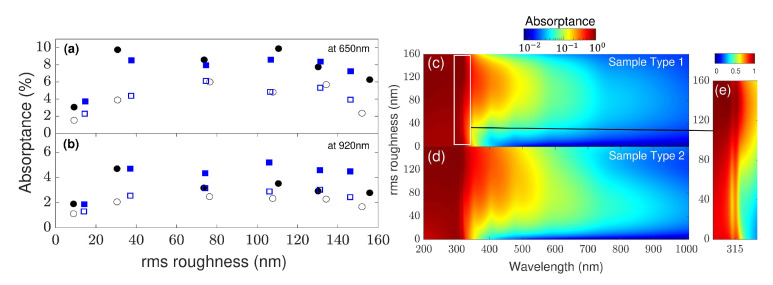
Absorptance as a function of rms roughness of surfaces for sample type 1 (Ag, filled black symbols) and sample type 2 (ZnO/Ag, empty black symbols) at incident wavelengths of (**a**) 650 nm and (**b**) 920 nm, respectively. The corresponding experimental data points extracted from Ref. [28] are shown in blue. (**c**,**d**) Absorptance on a logarithmic scale as a function of rms roughness and wavelength for the two sample types using a third-order polynomial fitting curve. (**e**) The plasmon peak of silver is observed around 315 nm. Here, a linear scale was used for clarity.

**Figure 6 nanomaterials-11-00113-f006:**
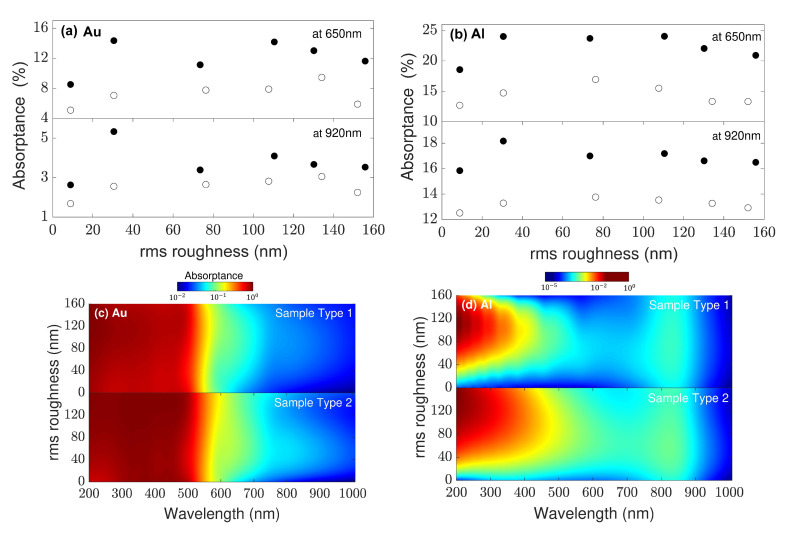
Same as in Figure 5, replacing the material function of silver with the corresponding functions for (**a**,**c**) gold and (**b**,**d**) aluminum.

**Table 1 nanomaterials-11-00113-t001:** The surface roughness of the different samples used in the calculations and the respective thickness of the Ag layer that best reproduces the experimental result.

Type 1 rms (nm)	Type 2 rms (nm)	Ag Thickness (nm)
8.86	8.86	65
30.48	30.48	85
76.34	73.51	105
107.69	110.48	195
134.17	130.19	255
151.97	155.81	315
0 (flat)	0 (flat)	315

**Table 2 nanomaterials-11-00113-t002:** Comparing the calculated rms roughness extracted from the original AFM images based on different numerical sources.

This Work	Gwyddion	Ref. [28]
9.30	9.34	14
30.48	30.44	37
76.34	76.21	74
107.69	107.50	106
134.17	133.90	131
151.97	151.60	146

## Data Availability

The datasets generated and analyzed during the presented study are available from the corresponding author on reasonable request.

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
