# Peer review of "Light Scattering from Rough Silver Surfaces: Modeling of Absorption Loss Measurements"

_nanomaterials, 2021, doi:10.3390/nano11010113_

Round 1
Reviewer 1 Report
This paper presents and interesting method to link sample roughness to its optical absorbance. However I think the following points coud be improved :
- for the sake of clarity, equation 6 could be given first and then equation 2 and 3 could be presented. This would make clearer the relationship between Haze function and roughness.
- In figure 3 there are case where transmission and absorbance are both close to 1 that is not possible according to equation 5
- the polynomial fits as a function of rouhness do not bring more information and could be misleading. I suggest to remove them.
Author Response
"Please see the attachment."

Reviewer 2 Report
Authors of the paper “Light scattering from rough silver surfaces: Modeling of absorption loss measurements” consider a model for the light loss in a case of rough metal surface. The paper contains new and interesting material, however, I propose a minor revision.
- My main recommendation is to rearrange the paper structure.
- a) I believe that the part of the Introduction that describes samples and their preparation (Ref [27] and Fig. 1) belong to a Methods section. Table 1 is first mentioned in the section Introduction, but is positioned in the Methods section. This only confirms my suggestion. OR move Table 1 to the section Introduction.
- b) Authors mention Table 2 in the Methodology section, but place it in the Appendix. This is not correct. Table 2 should be placed near the first mention of it.
- c) To me information placed as Method looks more like Results. it is obvious that authors have calculated it all on themselves, although Method section should describe already known methods and formulae. Or explain why this is still method, or include it into the Results and discussion section.
- d) Figs. 3 and 4 are placed in a Method section, but first mentioned in a Results and discussion section. This is not correct.
- e) Discussion from Appemdixes A and B to my opinion belongs to a Method section.
- I see some gaps in the reference list.
- a) DOI of references 8, 9, 12, 13, 15, 19, 20, 23, 24, 25, 32, 33, 35, 36, 38 is not specified, but these papers do have DOI. Please, add the DOI to these papers
- b) The authors cite paper [3] as an example of a photonic crystals use. However, I see that the paper [3] concerns collective charge oscillations in electrolytes. I believe that this reference is not relevant.
- c) The reference [5] conserns all-dielectric materials, however the authors cite it as a work dedicated to metasurfaces. Besides, this article is originally in Chinese, which should be stated in the reference list.
- d) The link to the Gwyddion, given as a footnote, should be made as a reference.
- e) The reference [26] is written by a lot of authors. I believe the form “ [26]. Hamed, T.A. et al. Multiscale in modelling and validation for solar photovoltaics. EPJ Photovoltaics 2018, 9, 10. doi:10.1051/epjpv/2018008.” will not cause disrespect.
- As far, as I have understood, Fig. 2 a are patterns taken from a paper [27]. Fig. 4 also contains data taken directly from Ref [27]. You need to get permission to reproduce the patterns and data. I see that the authors of ref [27] have obviously provided you with the data. However, I believe that the Nanomaterials editor will need an official permission.
